# Shear stress activates ADAM10 sheddase to regulate Notch1 via the Piezo1 force sensor in endothelial cells

Vincenza Caolo[1†], Marjolaine Debant[1†], Naima Endesh[1], T Simon Futers[1], Laeticia Lichtenstein[1], Fiona Bartoli[1], Gregory Parsonage[1], Elizabeth AV Jones[2*], David J Beech[1*]

[1]Leeds Institute of Cardiovascular and Metabolic Medicine, School of Medicine, University of Leeds, Leeds, United Kingdom; [2]Department of Cardiovascular Sciences, Centre for Molecular and Vascular Biology, Leuven, Belgium

**Abstract** Mechanical force is a determinant of Notch signalling but the mechanism of force detection and its coupling to Notch are unclear. We propose a role for Piezo1 channels, which are mechanically-activated non-selective cation channels. In cultured microvascular endothelial cells, Piezo1 channel activation by either shear stress or a chemical agonist Yoda1 activated a disintegrin and metalloproteinase domain-containing protein 10 (ADAM10), a $Ca^{2+}$-regulated transmembrane sheddase that mediates S2 Notch1 cleavage. Consistent with this observation, we found Piezo1-dependent increase in the abundance of Notch1 intracellular domain (NICD) that depended on ADAM10 and the downstream S3 cleavage enzyme, γ-secretase. Conditional endothelial-specific disruption of Piezo1 in adult mice suppressed the expression of multiple Notch1 target genes in hepatic vasculature, suggesting constitutive functional importance in vivo. The data suggest that Piezo1 is a mechanism conferring force sensitivity on ADAM10 and Notch1 with downstream consequences for sustained activation of Notch1 target genes and potentially other processes.

**\*For correspondence:**
liz.jones@kuleuven.be (EAVJ);
d.j.beech@leeds.ac.uk (DJB)

[†]These authors contributed equally to this work

**Competing interests:** The authors declare that no competing interests exist.

## Introduction

Mammalian Notch proteins were identified following studies in *D. melanogaster* that linked genetic abnormality to wing notch (*Siebel and Lendahl, 2017*). Extensive research then revealed major roles in the transfer of information between cells in health and disease (*Siebel and Lendahl, 2017*). Each of the four Notch receptors (Notch1-4) is a membrane protein that is trans coupled to a membrane-anchored ligand such as Deltalike 4 (DLL4). Though the initiation of Notch signalling is often considered to occur through ligand-receptor complex formation, mechanical force also plays an important role in this activation whereby a pulling force arising from ligand endocytosis causes trans activation (*Siebel and Lendahl, 2017*; *Gordon et al., 2015*). Furthermore it became apparent that frictional force from fluid flow also stimulates Notch1, but how this force couples to the Notch mechanism is unknown (*Fang et al., 2017*; *Mack et al., 2017*; *Lee et al., 2016*; *Jahnsen et al., 2015*). Therefore mechanical forces would seem to play key roles in Notch regulation. Further information is needed on how this is achieved.

Piezo1 channels are key players in the sensing of shear stress and lateral force applied to plasma membranes (membrane tension) (*Coste et al., 2010*; *Murthy et al., 2017*; *Li et al., 2014*; *Rode et al., 2017*; *Ranade et al., 2014*; *Wu et al., 2017*; *Maneshi et al., 2018*; *Wang et al., 2016*; *Beech and Kalli, 2019*). While there are multiple candidate sensors, Piezo1 channels are notable because of broad agreement amongst investigators that they are direct sensors of physiological force. As such they are now considered to be bona fide force sensors that evolved to sense and transduce force as a primary function (*Murthy et al., 2017*; *Wu et al., 2017*; *Beech and Kalli,*

*2019*). Piezo1 channels are exquisitely sensitive to membrane tension (*Lewis and Grandl, 2015*) and readily able to confer force-sensing capacity on cells that are otherwise poorly sensitive (*Coste et al., 2010*; *Li et al., 2014*). Reconstitution of Piezo1 channels in artificial lipid bilayers generates force-sensing channels (*Syeda et al., 2016*) and native Piezo1 channels in excised membrane patches respond robustly to mechanical force in the absence of intracellular factors (*Rode et al., 2017*).

Global knockout of Piezo1 in mice is embryonic lethal just after the heart starts to beat, apparently because of failed vascular maturation (*Li et al., 2014*; *Ranade et al., 2014*; *Beech and Kalli, 2019*). Particular functional significance is thought to arise in endothelial cells, where requirements in cell adherence, migration and proliferation and angiogenesis, wound closure, vascular permeability and blood pressure have been described (*Beech and Kalli, 2019*). Human genetic studies have suggested importance specifically in lymphatic vasculature (*Fotiou et al., 2015*) and varicose vein formation (*Fukaya et al., 2018*). Piezo1, like Notch1 (*Siebel and Lendahl, 2017*), is not restricted to endothelial cells or vasculature (*Murthy et al., 2017*; *Beech and Kalli, 2019*). There are also roles in erythrocytes and immune cells, neural stem cells, skeletal muscle cells, fibroblasts and many other cells and systems, as reviewed recently (*Beech and Kalli, 2019*).

Piezo1 channels are $Ca^{2+}$-permeable non-selective cationic channels, so when force causes them to open there is $Ca^{2+}$ entry, elevation of the cytosolic $Ca^{2+}$ concentration and regulation of $Ca^{2+}$-dependent mechanisms (*Coste et al., 2010*; *Murthy et al., 2017*). Potentially relevant to such a system is $Ca^{2+}$ and $Ca^{2+}$-calmodulin regulation of ADAM10 (*Nagano et al., 2004*; *Maretzky et al., 2015*), a metalloprotease or sheddase that catalyses rate-limiting S2 cleavage of Notch1 prior to γ-secretase-mediated S3 cleavage and release of Notch1 intracellular domain (NICD), driving downstream transcription (*Siebel and Lendahl, 2017*; *Alabi et al., 2018*; *Anders et al., 2001*). Therefore, we speculated about a relationship between Piezo1 and Notch1. We focussed on endothelial cells where both proteins are prominent and have established functional significance (*Siebel and Lendahl, 2017*; *Murthy et al., 2017*; *Li et al., 2014*; *Rode et al., 2017*; *Wu et al., 2017*; *Alabi et al., 2018*).

## Results

### Shear stress-induced S3 cleavage of Notch1 is Piezo1 dependent

The canonical pathway for Notch1 activation involves cleavage at the S3 site, which generates NICD, a protein of about 110 kDa that can be detected by western blotting. The pathway was previously suggested to be activated by shear stress (*Mack et al., 2017*). We first tested if we could reproduce the shear stress activation, using cultured human microvascular endothelial cells (HMVEC-Cs) as a model of endothelium. Laminar shear stress of 10 dyn.cm$^{-2}$ was applied to the cells for 1 hr and then abundance of NICD was measured. As expected, NICD was significantly increased (*Figure 1a, b*, *Figure 1—figure supplement 1*). We next validated a Piezo1-targeted siRNA for specific Piezo1 depletion (*Figure 1—figure supplements 2* and *3*). Strikingly, depletion of Piezo1 strongly reduced the amount of NICD in the shear stress condition, so much so that it became similar to that of the static control siRNA condition (*Figure 1a,b*). In some experiments there was unexpected reduction in NICD in the static (no shear stress) condition (*Figure 1a*) but accurate determination above non-specific background was technically challenging and the effect was not always evident. Although statistical analysis indicated no significant change in this basal NICD signal (*Figure 1b*), its existence in some individual experiments complicated our determination of whether shear stress induced an increase in NICD (*Figure 1a,b*). Nevertheless, in some individual experiments there was clearly no effect of shear stress on NICD after Piezo1 depletion (*Figure 1a*) and statistical analysis of all experiments confirmed no significant effect of shear stress (*Figure 1b*). The data suggest that Piezo1 is needed for normal elevation of NICD in shear stress and that it may be a factor regulating basal NICD.

### Chemical activation of Piezo1 also increases NICD abundance

The above data could be explained by an indirect role of Piezo1 or by Piezo1 as the starting point: that is the sensor of shear stress that triggers downstream changes. To test if Piezo1 can be the starting point, we circumvented shear stress and specifically activated Piezo1 chemically by using a

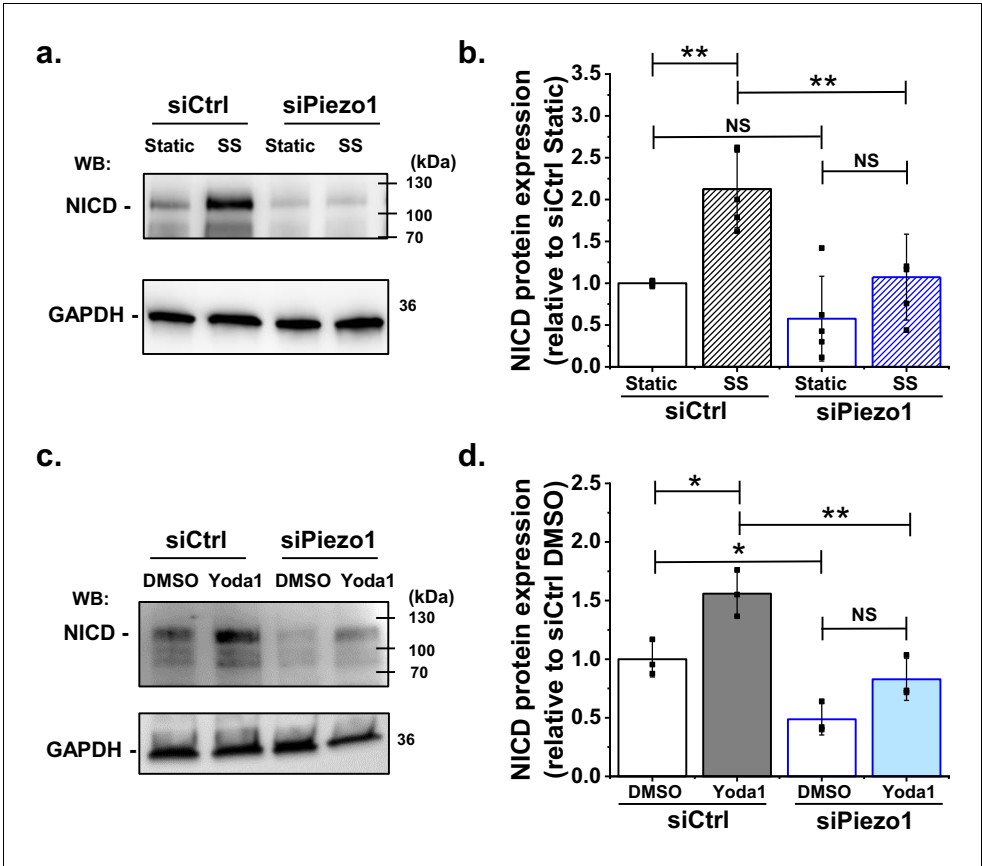

**Figure 1.** Shear stress induced S3 cleavage of Notch1 is Piezo1 dependent. (**a**) Representative Western blot labelled with anti-NICD and anti-GAPDH (loading control) antibodies for HMVEC-Cs exposed to 10 dyn.cm$^{-2}$ laminar shear stress (SS) for 1 hr. Static was without SS. Cells were transfected with control siRNA (siCtrl) or Piezo1 siRNA (siPiezo1). The expected mass of NICD is 110 kDa. Lower molecular bands were also apparent in some experiments and may have been degraded NICD. (**b**) Quantification of data of the type exemplified in (**a**), showing mean ± SD data for abundance of NICD normalized to siCtrl Static (n = 4). (**c**) Representative Western blot labelled with anti-NICD and anti-GAPDH antibodies for HMVEC-Cs treated for 30 min with 0.2 μM Yoda1 or vehicle (DMSO) after transfection with control siRNA (siCtrl) or Piezo1 siRNA (siPiezo1). (**d**) Quantification of data of the type exemplified in (**c**), showing mean ± SD for abundance of NICD normalized to siCtrl DMSO (n = 3). Statistical analysis: Two-way ANOVA test was used, indicating *p<0.05, **p<0.01 or not significantly different (NS). The online version of this article includes the following source data and figure supplement(s) for figure 1:

**Source data 1.** Source data for *Figure 1*.
**Figure supplement 1.** Uncropped western blots for *Figure 1a* and *Figure 1c*.
**Figure supplement 2.** Additional supporting data for *Figure 1*.
**Figure supplement 2—source data 1.** Source data for *Figure 1—figure supplement 2*.
**Figure supplement 3.** Uncropped western blots for the *Figure 1—figure supplement 2b*.
**Figure supplement 4.** The response depends on γ-secretase.
**Figure supplement 4—source data 1.** Source data for *Figure 1—figure supplement 4*.
**Figure supplement 5.** Uncropped western blots for *Figure 1—figure supplement 4a*.

synthetic small-molecule agonist (Yoda1). Yoda1 is described to enhance the force sensitivity of Piezo1 channels (*Syeda et al., 2015*; *Lacroix et al., 2018*; *Wang et al., 2018*; *Evans et al., 2018*). There is inherent force in cell membranes and we previously showed that Yoda1 activates Piezo1 in endothelial cells without the need for applied exogenous force (*Evans et al., 2018*). Therefore we applied Yoda1 at 0.2 μM, the concentration previously reported for half-maximal activation of native endothelial Piezo1 channels (*Evans et al., 2018*). Strikingly, in static conditions, Yoda1 alone could stimulate increased NICD (*Figure 1c,d*). Piezo1 siRNA suppressed the Yoda1 effect (*Figure 1c,d*). In these experiments, the effect of Piezo1 to reduce basal NICD was statistically significant (*Figure 1d*).

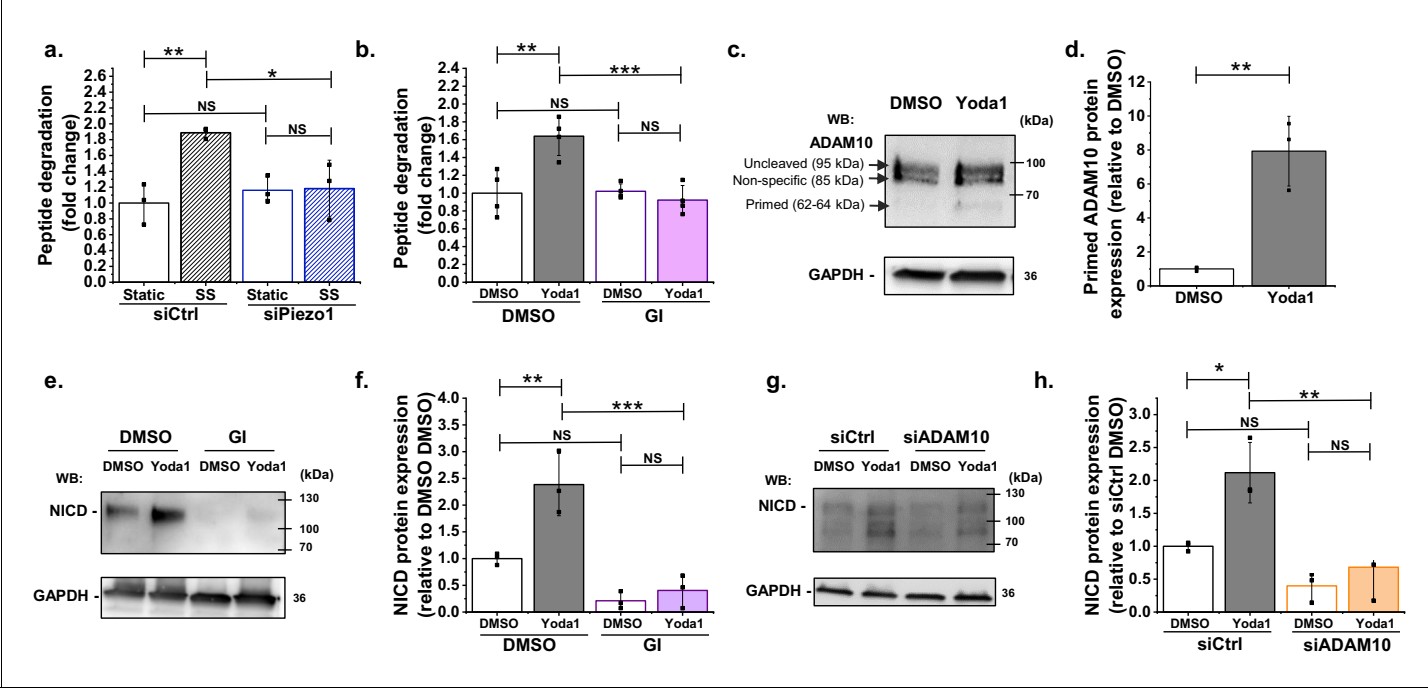

**Figure 2.** ADAM10 is important for Piezo1 regulation of NICD. (**a**) ADAM10 enzyme activity assessed by specific peptide degradation and subsequent fluorescence emission after 30 min exposure of HMVEC-Cs to 10 dyn.cm$^{-2}$ laminar shear stress (SS). Static was without SS. Cells were transfected with control siRNA (siCtrl) or Piezo1 siRNA (siPiezo1). Data are shown as mean ± SD data (n = 3) relative to static condition. (**b**) ADAM10 enzyme activity assessed after 30 min treatment of HMVEC-Cs with 0.2 μM Yoda1 in the absence or presence of 5 μM GI254023X (GI). Data are shown as mean ± SD data (n = 4) relative to DMSO condition. (**c, d**) Quantification of uncleaved (95 kDa) and cleaved (62–64 kDa) ADAM10 in HMVEC-Cs after treatment for 30 min with Yoda1 (0.2 μM). The 85 kDa band between the uncleaved and cleaved ADAM10 was non-specific labelling not related to ADAM10 (*Figure 2—figure supplement 2*). Data represent mean ± SD (n = 3) and normalization was to the reference protein, GAPDH. (**e**) Example Western blot labelled with anti-NICD and anti-GAPDH antibodies for HMVEC-Cs treated for 30 min with 0.2 μM Yoda1 or vehicle (DMSO) in the absence or presence of 5 μM GI254023X (GI). (**f**) Quantification of data of the type exemplified in (**e**), showing mean ± SD data for abundance of NICD normalized to DMSO (n = 3). (**g**) Representative Western blot labelled with anti-NICD and anti-GAPDH antibodies for HMVEC-Cs treated for 30 min with 0.2 μM Yoda1 or vehicle (DMSO) after transfection with control siRNA (siCtrl) or ADAM10 siRNA (siADAM10). (**h**) Quantification of data of the type exemplified in (**g**), showing mean ± SD data for abundance of NICD normalized to siCtrl DMSO (n = 3). Statistical analysis: Two-way ANOVA test was used for (**a, b, f, h**), indicating *p<0.05, **p<0.01, ***p<0.001; t-Test was used for (**d**), indicating **p<0.01; NS, not significantly different.

The online version of this article includes the following source data and figure supplement(s) for figure 2:

**Source data 1.** Source data for *Figure 2*.
**Figure supplement 1.** Uncropped western blots for *Figure 2c*, *Figure 2e* and *Figure 2g*.
**Figure supplement 2.** Additional supporting data for *Figure 2*.
**Figure supplement 2—source data 1.** Source data for *Figure 2—figure supplement 2*.
**Figure supplement 3.** Uncropped western blots for *Figure 2—figure supplement 2b* and *Figure 2—figure supplement 2i*.

The data support the hypothesis that Piezo1 is the sensor for shear stress that then triggers downstream Notch1 processing. The data also suggest a constitutive role of Piezo1 in maintaining a basal level of Notch1 cleavage.

## γ-secretase is required

S3 cleavage is mediated by γ-secretase (*Siebel and Lendahl, 2017*). Therefore we tested the role of γ-secretase by treating cells with 10 μM DAPT (N-[N-(3,5-difluorophenacetyl)-l-alanyl]-S-phenylglycine t-butylester), a commonly used γ-secretase inhibitor (*Mack et al., 2017*; *Imbimbo, 2008*). DAPT had an effect on NICD that was similar to that of Piezo1 siRNA, reducing basal NICD and ablating the ability of Yoda1 to increase NICD (*Figure 1—figure supplements 4* and *5*). Although DAPT inhibited Yoda1-evoked Ca$^{2+}$ entry by about 30%, such a potentially non-specific effect was unlikely

to have been sufficient to explain its effect on NICD (*Figure 1—figure supplements 4* and *5*). The data suggest that Piezo1-mediated and constitutively-generated NICD require γ-secretase.

## There is Piezo1-dependent and Piezo1-mediated activation of the S2 cleavage enzyme, ADAM10

In the canonical Notch1 pathway, S2 cleavage is required prior to S3 cleavage. A mediator of S2 cleavage is ADAM10 (*Siebel and Lendahl, 2017*; *Alabi et al., 2018*). Therefore we measured ADAM10 enzymatic activity (*Figure 2a*). Importantly, even after only 30 min shear stress, there was significant increase in ADAM10 activity and this effect was abolished by Piezo1 depletion (*Figure 2a*). Similarly, in static conditions, Yoda1 activated ADAM10, again consistent with Piezo1 being the starting point (*Figure 2b*). The effect was prevented by the widely used ADAM10 inhibitor GI254023X ((2R,3S)−3-(Formyl-hydroxyamino)−2-(3-phenyl-1-propyl) butanoic acid[(1S)−2,2-dimethyl-1-methylcarbamoyl-1-propyl] amide) which is thought to act via the catalytic site (*Ludwig et al., 2005*; *Figure 2b*). We also quantified the abundance of ADAM10's cleaved form, a 62–64 kDa protein that is generated by proprotein convertase to enable enzymatic activity (*Anders et al., 2001*). The majority of ADAM10 was in the uncleaved form, a protein of 95 kDa that is inactive (*Figure 2c*, *Figure 2—figure supplements 1* and *2*). Yoda1 significantly increased the abundance of the cleaved form (*Figure 2c,d*, *Figure 2—figure supplement 1*). The data suggest that shear stress causes Piezo1-dependent activation of ADAM10 and that Piezo1 activation alone is sufficient to activate ADAM10.

## ADAM10 activation is required for the NICD effect

To investigate if ADAM10 activity is required for Piezo1 coupling to Notch1, we first tested the effect of the ADAM10 inhibitor, GI254023X. This agent strongly inhibited the ability of Yoda1 to increase NICD and suppressed basal NICD (*Figure 2e,f*). In these experiments we used 5 μM GI254023X. A 10-fold lower concentration of GI254023X (500 nM) also inhibited the Yoda1 effect on NICD (*Figure 2—figure supplement 2*), consistent with its nanomolar potency against ADAM10 (*Ludwig et al., 2005*). GI254023X (5 μM) had no effect on Yoda1-evoked $Ca^{2+}$ entry, suggesting that it did not act non-specifically (*Figure 2—figure supplements 2* and *3*). To independently test the role of ADAM10 we developed specific ADAM10 depletion by ADAM10-targeted siRNA (*Figure 2—figure supplement 2*). The effect of this ADAM10 depletion was similar to that of GI254023X (*Figure 2g,h cf Figure 2e,f*; *Figure 2—figure supplement 1*), supporting the hypothesis that ADAM10 is between Piezo1 and Notch1. The data suggest that Piezo1-mediated stimulation of ADAM10 enzyme activity is necessary for basal and stimulated effects on Notch1 cleavage.

## An ion pore blocker of Piezo1 inhibits ADAM10 activation

Activation of ADAM10 has been suggested to be mediated by $Ca^{2+}$ (*Nagano et al., 2004*; *Maretzky et al., 2015*). Therefore, the ability of Piezo1 to activate ADAM10 could be due to the ion channel property of Piezo1, which allows influx of cations such as $Ca^{2+}$ and $Na^+$ in response to mechanical activation (*Coste et al., 2010*; *Wu et al., 2017*). To test this mechanism experimentally, we used $Gd^{3+}$ (30 μM), which is a blocker of the Piezo1 channel pore (*Coste et al., 2010*). Importantly, $Gd^{3+}$ inhibited the ability of Yoda1 to activate ADAM10, consistent with the hypothesis that ion permeation through Piezo1 channels is critical for ADAM10 activation (*Figure 2—figure supplement 2*). $Gd^{3+}$ is not specific to Piezo1 channels but better agents are not currently known. The data suggest that the ion channel property of Piezo1 is critical in ADAM10 activation.

## Piezo1 activation regulates Notch1 target genes

An implication of Piezo1 causing S2 and S3 cleavage of Notch1 is that the expression of Notch1 target genes should also be activated. Therefore we quantified Notch1-regulated gene expression, focussing initially on the *HES1* gene which is Notch1- and flow- regulated (*Mack et al., 2017*), *DLL4* which is itself Notch1 regulated (*Caolo et al., 2010*) and *HEY1*, another Notch1 target gene (*Caolo et al., 2011*). Yoda1 caused striking increases in the expression of *HES1*, *DLL4* and *HEY1* genes (*Figure 3*). These effects of Yoda1 were suppressed by Piezo1 siRNA (*Figure 3a,b,c*), DAPT (*Figure 3d,e,f*), ADAM10 siRNA (*Figure 3g,h,i*) and GI254023X (*Figure 3j,k,l*). Expression of another Notch1 target gene, *HEY2* (*Lobe, 1997*), also appeared to be stimulated by Yoda1 but statistical

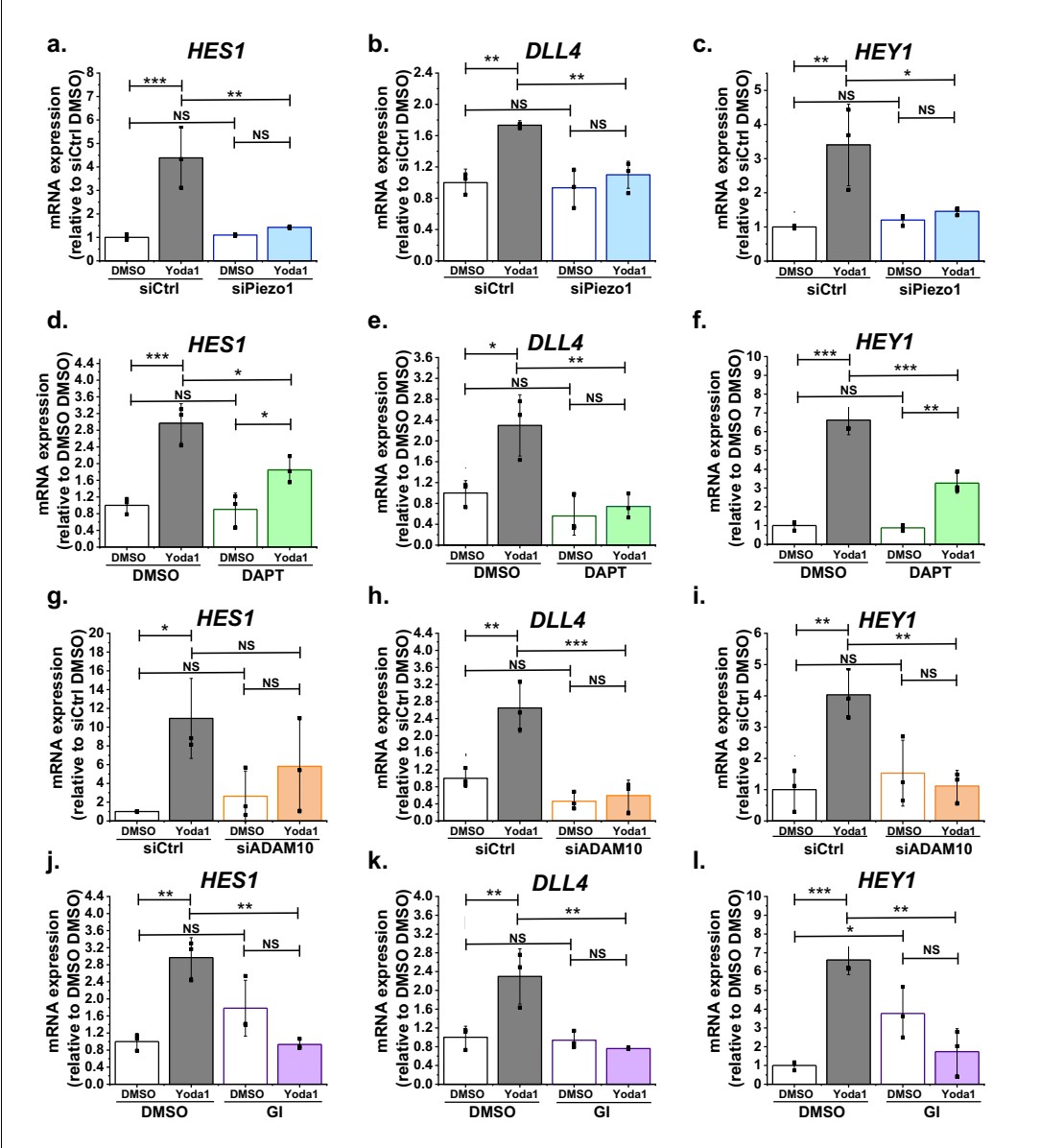

**Figure 3.** Function significance for downstream gene expression. (a, b, c) Summarized mean ± SD (n = 3) quantitative PCR data for fold-change in *HES1* (a), *DLL4* (b) and *HEY1* (c) mRNA in HMVEC-Cs treated for 2 hr with 0.2 μM Yoda1 or vehicle (DMSO) after transfection with control siRNA (siCtrl) or Piezo1 siRNA (siPiezo1). (d, e, f) Summarized mean ± SD (n = 3) quantitative PCR data for fold-change in *HES1* (d), *DLL4* (e) and *HEY1* (f) mRNA in HMVEC-Cs treated for 2 hr with 0.2 μM Yoda1 in the absence or presence of 10 μM DAPT. (g, h, i) Summarized mean ± SD (n = 3) quantitative PCR data for fold-change in *HES1* (g), *DLL4* (h) and *HEY1* (i) mRNA in HMVEC-Cs treated for 2 hr with 0.2 μM Yoda1 or vehicle (DMSO) after transfection with control siRNA (siCtrl) or ADAM10 siRNA (siADAM10). (j, k, l) Summarized mean ± SD (n = 3) quantitative PCR data for fold-change in *HES1* (j), *DLL4* (k) and *HEY1* (l) mRNA in HMVEC-Cs treated for 2 hr with 0.2 μM Yoda1 in the absence or presence of 5 μM GI254023X (GI). Normalization and statistical analysis: mRNA expression was normalized to *GAPDH* mRNA abundance. Two-way ANOVA test was used, indicating $*p<0.05$, $**p<0.01$, $***p<0.001$ or not significantly different (NS).

The online version of this article includes the following source data and figure supplement(s) for figure 3:

**Source data 1.** Source data for *Figure 3*.
**Figure supplement 1.** Additional data relating to *Figure 3*.
**Figure supplement 1—source data 1.** Source data for *Figure 3—figure supplement 1*.

significance was not achieved due to high variability in the response (*Figure 3—figure supplement 1*). Two other previously suggested Notch1 target genes (*HEY2* **Fischer et al., 2004** and *JAG1* **Foldi et al., 2010**) were not significantly affected, suggesting selective effects on certain Notch1

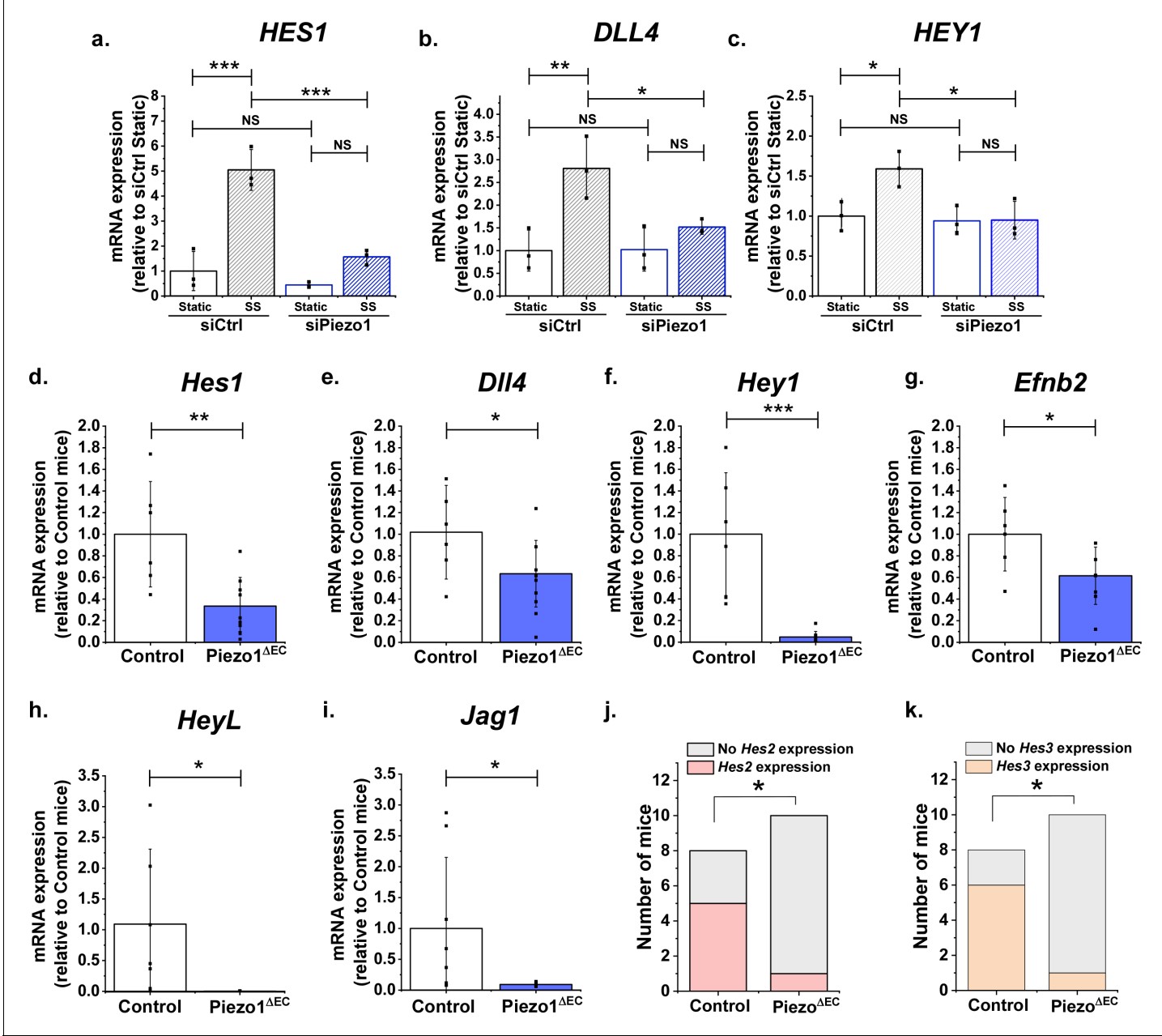

**Figure 4.** Endothelial Piezo1 is required for the gene expression of Notch1 targets in HMVEC-C exposed to shear stress and in mouse liver endothelial cells. (a, b, c) Summarized mean ± SD (n = 3) quantitative PCR data for fold-change in *HES1* (a), *DLL4* (b) and *HEY1* (c) mRNA in HMVEC-Cs exposed to 10 dyn.cm$^{-2}$ laminar shear stress (SS) for 2 hr after transfection with control siRNA (siCtrl) or Piezo1 siRNA (siPiezo1). *Hes1* (d), *Dll4* (e), *Hey1* (f) and *Efnb2* (g) mRNA expression in liver endothelial cells freshly-isolated from control mice (Control, n = 6) and endothelial Piezo1 knockout mice (Piezo1$^{\Delta EC}$) (n = 9). *HeyL* (h) (Control mice, n = 7; Piezo1$^{\Delta EC}$ mice, n = 7) and *Jag1* (i) (Control mice, n = 8; Piezo1$^{\Delta EC}$ mice, n = 9) mRNA expression in liver endothelial cells. Normalization and Statistical analysis: mRNA expression was normalized to abundance of *Actb* mRNA, which was not different between Control and Piezo1$^{\Delta EC}$ (*Figure 4—figure supplement 2*). *Hes2* (j) and *Hes3* (k) mRNA expression in liver endothelial cells freshly-isolated from control mice (Control, n = 8) and Piezo1$^{\Delta EC}$ mice (n = 10), represented as the number of mice with detectable expression or no detectable expression of the gene. Statistical analysis: Two-way ANOVA test was used for (a, b, c), indicating *p<0.05, **p<0.01, ***p<0.001. t-Test was used for (d, e, f, g, h, i) indicating significant difference of Piezo1$^{\Delta EC}$ cf Control *p<0.05, **p<0.01, ***p<0.01. Fisher's exact test was used for (j, k) indicating significant difference of Piezo1$^{\Delta EC}$ cf Control *p<0.05. NS indicates not significantly different.

The online version of this article includes the following source data and figure supplement(s) for figure 4:

**Source data 1.** Source data for *Figure 4*.
**Figure supplement 1.** Additional data relating to *Figure 4*.
**Figure supplement 1—source data 1.** Source data for *Figure 4—figure supplement 1*.
*Figure 4 continued on next page*

*Figure 4 continued*

**Figure supplement 2.** Supporting data for *Figure 4*.
**Figure supplement 2—source data 1.** Source data for *Figure 4—figure supplement 2*.
**Figure supplement 3.** Notch1 target gene expression changes are not evident in whole liver.

target genes (*Figure 3—figure supplement 1*). Expression of two other potential targets, *HES3* and *HEYL*, was not reliably detected and so effects of Yoda1 could not be determined. The data suggest that Piezo1 signalling via ADAM10 and γ-secretase to Notch1 and NICD is functionally important for downstream gene regulation.

## Piezo1-dependent regulation of Notch1 target genes by shear stress

The above data show that Piezo1 can activate Notch1 target genes but do not show that it is relevant to shear stress regulation of these genes. Therefore, we also investigated the effect of shear stress on *HES1, DLL4 and HEY1* expression. As expected, shear stress upregulated the expression of all three genes (*Figure 4a–c*). Importantly, all of these effects were Piezo1-dependent (*Figure 4a–c*). There was a trend towards similar regulation of *HES2* but expression of *HEY2* and *JAG1* was unaffected (*Figure 4—figure supplement 1*). The data suggest that shear stress coupling to Notch1 target genes is mediated by Piezo1.

## Endothelial Piezo1 is required for Notch1 target gene expression in mice

The above findings suggest that endothelial Piezo1 might be important for Notch1 target gene expression in vivo. Therefore, we conditionally disrupted Piezo1 specifically in endothelium of adult mice (Piezo1$^{\Delta EC}$ mice), as previously described (*Rode et al., 2017*). Two weeks after disruption in vivo, endothelial cells were isolated and gene expression was measured acutely to reflect the normal Notch1 target gene expression in the endothelial cells of the mice. We elected to study hepatic microvascular endothelial cells because these cells were previously demonstrated to contain functional Piezo1 channels (*Rode et al., 2017*) and Notch1 has known relevance in liver endothelial sinusoids (*Alabi et al., 2018; Cuervo et al., 2016*). Importantly, expression of *Hes1, Dll4* and *Hey1* genes were all found to be downregulated in the Piezo1$^{\Delta EC}$ condition (*Figure 4d–f*). Moreover, in contrast to some of our findings in HMVEC-Cs, there was similar downregulation of 5 other Notch1 target genes, including *Efnb2*, which is another Notch1- and flow-regulated gene (*Mack et al., 2017; Jahnsen et al., 2015; Figure 4g–k*). *Piezo1* gene expression was confirmed as depleted in these endothelial cells from Piezo1$^{\Delta EC}$ mice, as expected, whereas expressions of the reference gene, *Actb*, and endothelial marker gene, *Tek*, were unaffected, suggesting specificity (*Figure 4—figure supplement 2*). We were unable to detect *Hey2* expression in these endothelial cells. In contrast to the findings in isolated endothelial cells, whole liver showed no significant changes in expression of *Hes1, Dll4, Hey1, Efnb2, HeyL* or *Jag1* in Piezo1$^{\Delta EC}$ mice, consistent with the effects being restricted to the endothelial cell population (*Figure 4—figure supplement 3*). Expression of *Hes2* and *Hes3* could not be detected in whole liver samples. The data suggest that endothelial Piezo1 is normally required for physiological Notch1 target gene expression in vivo.

## Discussion

The study has identified a connection between Piezo1 channels and Notch1 signalling and thus a novel mechanism by which Notch1 can be regulated and impacted by mechanical force. Based on our data we propose a pathway in which activation of Piezo1 channels leads to stimulation of ADAM10 for S2 cleavage of Notch1, which then enables intracellular S3 cleavage of Notch1 by γ-secretase and release of NICD for association with transcriptional regulators such as RBPJ and the control of multiple Notch1 target genes (*Figure 5*). In this way, Notch1 is coupled to an exceptional force sensor, the Piezo1 channel. Other mechanisms by which Notch1 achieves force sensitivity are not excluded but we suggest that Piezo1 is a mechanism of biological significance because we found that endothelial-specific disruption of Piezo1 in vivo disturbed Notch1-regulated gene expression in

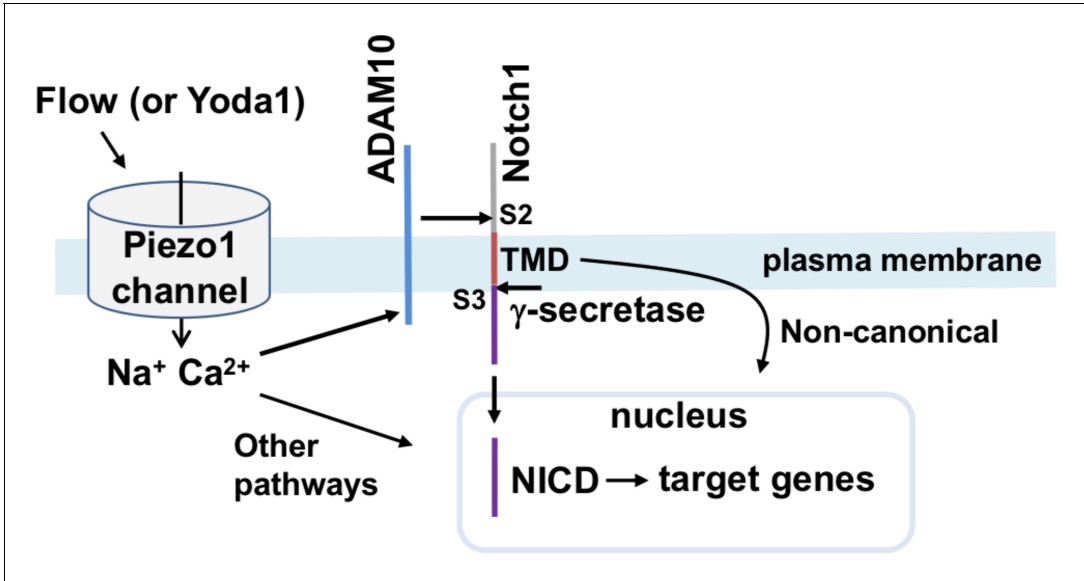

**Figure 5.** Summary of the proposed pathway. Activation of Piezo1 channel by mechanical force (e.g. fluid flow) or chemical agonist (Yoda1) causes elevation of the intracellular $Ca^{2+}$ concentration which stimulates enzymatic activity of ADAM10 to cause S2 and S3 Notch1 cleavage and release of NICD to drive target gene expression that includes increased expression of Hes1, Dll4, Hey1, HeyL, Jag1, Hes2, Hes3 and Efnb2 in hepatic vasculature of the mouse. In the schematic we also include the suggested contributions from ADAM10/Notch1/NICD independent signalling from Piezo1 and non-canonical Notch1 signalling via the Notch1 transmembrane domain (TMD), as described and referenced in the text.

hepatic endothelial cells, a site known to be Notch-regulated. Our findings are consistent with prior studies of *D melanogaster* that inferred Notch to be downstream of Piezo (*He et al., 2018*) and so the proposed mechanism may have broad relevance.

The pathway proposed in *Figure 5* is based primarily on evidence from our studies of human endothelial cells in culture (HMVEC-Cs) but we also investigated the relevance to endothelial cells in vivo by studying endothelium freshly isolated from whole liver of control and endothelial Piezo1 knockout (Piezo1$^{\Delta EC}$) mice. These hepatic experiments show that loss of Piezo1 in endothelium leads to reduced expression of multiple Notch1 target genes in endothelium. Therefore, the data suggest that Piezo1 positively impacts Notch1 target genes in vivo and that the pathway of *Figure 5* is active in vivo. There were intriguing differences in the Notch1 target genes affected by Piezo1 in HMVEC-Cs compared with freshly-isolated mouse endothelium, with a greater range affected in the mouse cells. Such data suggest context-dependent regulation of Notch1 target genes by Piezo1 or endothelial cell heterogeneity. Piezo1 activates not only ADAM10/Notch1 signalling but also calpain (*Li et al., 2014*), eNOS (*Li et al., 2014*; *Wang et al., 2016*) and other signalling mechanisms (*Beech and Kalli, 2019*; *Albarrán-Juárez et al., 2018*). We speculate that these other pathways may also impact Notch1 target genes directly or indirectly, either to prevent or enhance the positive impact of Piezo1 via ADAM10 and Notch1.

The specific downstream implications of Piezo1-ADAM10/Notch1 signalling in the liver remain to be explicitly determined but we know that prior work showed important effects of endothelial-specific disruption of Notch1-regulated RBPJ on hepatic microvasculature (*Alabi et al., 2018*; *Cuervo et al., 2016*). Disruption of RBPJ in 6 week-old mice led to enlarged sinusoids between portal and central venules, and disruption earlier in postnatal development caused poor perfusion, hypoxia and liver necrosis (*Cuervo et al., 2016*). Therefore our observed requirement for Piezo1 in Notch1-regulated gene expression of hepatic endothelium suggests a positive role for the proposed Piezo1-ADAM10/Notch1 partnership in hepatic function. Moreover, roles of Piezo1 and Notch1 in chemotactic chemokine release, portal hypertension and liver fibrogenesis were recently suggested (*Hilscher et al., 2019*) and will be interesting to investigate further in regard to shear stress activation of Piezo1-ADAM10/Notch1 signalling. It is important to emphasise again, however, that Piezo1

signalling is extensive (*Beech and Kalli, 2019*) and therefore unlikely to be limited to ADAM10/Notch1. Therefore, studies of Piezo1$^{\Delta EC}$ mice will not inform specifically about the role of the Piezo1-ADAM10/Notch1 pathway. We will be able to determine the specific significance in vivo only if we first determine how to specifically disrupt this pathway relative to others.

The signalling pathway linking Piezo1 to ADAM10 is not precisely known and may involve yet to be identified components. We suggest that ion permeation through the Piezo1 channels is a critical starting point because Gd$^{3+}$, a blocker of the Piezo1 channel ion pore (*Coste et al., 2010*), prevented Yoda1 from stimulating ADAM10 activity. This suggests that cation flux is important. We suggest that it might be Ca$^{2+}$ influx (*Figure 5*) because Ca$^{2+}$ is an especially important intracellular messenger and prior work has shown Ca$^{2+}$ and Ca$^{2+}$-calmodulin-regulated mechanisms for S2 cleavage of Notch1 (*Nagano et al., 2004*; *Maretzky et al., 2015*), but signalling mediated by Na$^{+}$ influx has also been suggested (*Rode et al., 2017*) and cannot yet be excluded. It should be emphasised that the ADAM10 activity was measured after 30 min exposure to Yoda1, which is longer than the time required for maximum intracellular Ca$^{2+}$ elevation in response to Yoda1 (1 or 2 min). This time difference may reflect limited sensitivity of the ADAM10 activity assay compared with the intracellular Ca$^{2+}$ assay, which is known to be highly sensitive. Ca$^{2+}$ sensitivity of ADAM10 creates the potential for its regulation by various Ca$^{2+}$ entry and Ca$^{2+}$ release mechanisms. Piezo1 may, therefore, not be unique as a Ca$^{2+}$ entry mechanism regulating ADAM10 unless it has privileged spatial relationship with it, which is currently unknown.

Our observation of shear stress activation of ADAM10 via Piezo1 is consistent with previous work describing activation of ADAM10 by shear stress in human platelets (*Facey et al., 2016*), which also express Piezo1 (*Ilkan et al., 2017*). The finding has implications for Notch1 signalling but also more widely because ADAM10 is a sheddase that also targets E- and VE-cadherin, CD44 and cellular prion protein (*Nagano et al., 2004*; *Maretzky et al., 2005*; *Schulz et al., 2008*; *Jarosz-Griffiths et al., 2019*). Therefore its shear stress regulation via Piezo1 may be broadly relevant, for example in adherens junction biology and cartilage integrity, where functional importance of Piezo1 is already suggested (*Friedrich et al., 2019*; *Lee et al., 2014*), and amyloid plaque formation, where Piezo1 was originally detected (*Satoh et al., 2006*) and has been suggested to have functional importance in combination with *E coli* infection (*Velasco-Estevez et al., 2018*).

A challenging aspect of our HMVEC-C experiments was the tendency for the inhibition or depletion of ADAM10/Piezo1 to reduce the constitutive abundance of NICD. This created a technical limitation because the basal abundance of NICD in static conditions sometimes became difficult to distinguish from non-specific background signals, leading the reference point for any effect of shear stress to be different and potentially unreliable. In some data sets the impression can be gained of an approximate doubling of the abundance of NICD in the ADAM10/Piezo1-depleted or inhibited condition. This could suggest ADAM10/Piezo1-independent stimulation of Notch1 by shear stress, a possibility that we do not exclude, but it is important to recognise the following: (1) Any such effect of shear stress was small relative to the robust effect seen in control conditions; (2) The total abundance of NICD in the shear stress ADAM10/Piezo1-inhibited or depleted condition was only about the same as (or less than) that of static control conditions; (3) No apparent ADAM10/Piezo1-independent effect of shear stress was validated by statistical significance; and (4) In some experiments there was no change in basal signal and yet the effect of shear stress was abolished by Piezo1 depletion (ADAM10 activity and *HEY1* gene expression). Therefore we conclude that Piezo1 is the dominant mechanism for shear stress activation of ADAM10/Notch1 in HMVEC-Cs and suggest that this effect is relevant in vivo based on our studies of hepatic endothelium of Piezo1$^{\Delta EC}$ mice.

It is unclear if the basal effect of Piezo1 on NICD in some experiments (*Figure 1a,d*) is functionally important or related to mechanical force. Our data suggest lack of relevance to the Notch1-regulated genes investigated in HMVEC-Cs, although there was a hint of an effect in some experiments on *HES1* expression (*Figure 4a*). There are several possible explanations for a basal effect on NICD because multiple signalling pathways are activated by Piezo1 (*Beech and Kalli, 2019*). However, our favoured hypothesis is that basal mechanical activation of Piezo1 occurs via physical interaction of the cells with the substrate, which could explain the variability in the effect depending on the type of experiment. Previous work has shown that increasing substrate stiffness activates Piezo1 (*Pathak et al., 2014*). Another interesting possibility is that basal regulation, and indeed shear stress-regulation, depends on relationships of Piezo1 (*Nourse and Pathak, 2017*) and Notch1

(*Polacheck et al., 2017*) to cytoskeleton that depend on how cells interact with each other and the substrate. We previously showed that Piezo1 depletion affects cytoskeletal structure in endothelial cells (*Li et al., 2014*) and so there is a possibility that this plays a role in the interplay between Piezo1 and Notch1.

The pathway proposed (*Figure 5*) focusses on evidence from our study but, in addition, there may be relevance to non-transcriptional signalling through non-canonical exposure of the Notch1 transmembrane domain (TMD) to promote the formation of a complex of LAR phosphatase, TRIO guanidine-exchange factor and VE-cadherin and thus regulate barrier function (*Polacheck et al., 2017*) where Piezo1 has important roles (*Friedrich et al., 2019*; *Zhong et al., 2020*). Piezo1 signalling would be expected to increase availability of the TMD but this remains to be tested experimentally.

A prior report has suggested lack of specificity of Yoda1 for Piezo1 channels (*Dela Paz and Frangos, 2018*) but other prior studies have shown that genetic deletion of Piezo1 abolishes Yoda1's effects (*Rode et al., 2017*; *Friedrich et al., 2019*; *Cahalan et al., 2015*; *Romac et al., 2018*) and that Piezo1 knockdown by RNA interference suppresses its effects (*Wang et al., 2016*; *Albarrán-Juárez et al., 2018*; *Nonomura et al., 2018*; *Iring et al., 2019*; *Tsuchiya et al., 2018*; *Liu et al., 2018*), consistent with Yoda1 having only Piezo1-mediated effects. Specific structure-activity requirements of Yoda1 at Piezo1 have been observed (*Evans et al., 2018*) and Yoda1 does not activate Piezo2, the only other Piezo channel (*Syeda et al., 2015*). Here we observed that Piezo1-specific siRNA suppressed or abolished Yoda1 effects. Although it is important to seek Piezo1-specific activation with agents such as Yoda1 (because mechanical force is not specific to Piezo1), such agents do not necessarily mimic activation by a physiological factor. Importantly, therefore, we showed that Yoda1 indeed mimicked the effect of laminar shear stress because shear stress similarly activated ADAM10, increased the abundance of NICD and stimulated Notch1 target gene expression in a Piezo1-dependent manner. Moreover, genetic disruption of endothelial Piezo1 in vivo, in the absence of Yoda1, led to loss of Notch1 signalling. Therefore Piezo1 regulation of the ADAM10/Notch1 pathway is physiological and not dependent on Yoda1.

In conclusion, we connect the Piezo1 mechanosensing ion channel with the extensive prior discoveries of the ADAM and Notch fields (*Siebel and Lendahl, 2017*; *Alabi et al., 2018*). The ability of Piezo1 to activate ADAM10 and Notch1 suggests that these mechanisms are downstream of Piezo1 and therefore linked via Piezo1 to changes in physiological force. There are likely to be broad-ranging implications for hepatic biology, as we suggest, but also for other biology linked to ADAM10 and Notch1 (*Siebel and Lendahl, 2017*; *Alabi et al., 2018*). Understanding how to specifically disrupt the Piezo1-ADAM10/Notch1 partnership will be important if we are to better decipher the roles of the pathway and explore its therapeutic potential.

# Materials and methods

**Key resources table**

| Reagent type (species) or resource | Designation | Source or reference | Identifiers | Additional information |
|---|---|---|---|---|
| Strain, strain background (*Mus musculus*, C57BL6/J, male) | Piezo1$^{flox/flox}$/Cdh5-Cre | University of Leeds (*Rode et al., 2017*) | N/A | |
| Cultured cells (*Homo sapiens*) | HMVEC-Cs | Lonza | Cat# CC-7030 | |
| Antibody | anti-cleaved Notch1 val1744 (D3B8) (Rabbit monoclonal) | Cell Signaling Technology | Cat# 4147, RRID:AB_2153348 | WB (1:1000) |
| Antibody | anti-ADAM10 (Rabbit polyclonal) | Merck Millipore | Cat# AB19026, RRID:AB_2242320 | WB (1:1000) |
| Sequence-based reagent | ON-TARGET plus Control siRNA | Dharmacon | Cat# L-001810 | |

*Continued on next page*

*Continued*

| Reagent type (species) or resource | Designation | Source or reference | Identifiers | Additional information |
|---|---|---|---|---|
| Sequence-based reagent | ON-TARGET plus SMARTpool Human siRNA ADAM10 | Dharmacon | Cat# L-004503 | |
| Sequence-based reagent | Piezo1 siRNA | Sigma-Aldrich | N/A | GCAAGUUCGUGCGCGGAUU[DT][DT] |
| Commercial assay or kit | SensoLyte 520 ADAM10 Activity Assay Kit | AnaSpec Inc | Cat# AS-72226 | Use kit directions |
| Chemical compound, drug | GI 254023X | Tocris Bioscience | Cat# 3995 | |
| Chemical compound, drug | DAPT | Sigma-Aldrich | Cat# D5942 | |
| Chemical compound, drug | Yoda1 | Tocris Bioscience | Cat# 5586–10 | |

## Piezo1 mutant mice

All animal use was authorized by the University of Leeds Animal Ethics Committee and Home Office UK (Project Licence P606320FB to David J Beech). Genotypes were determined using real-time PCR with specific probes designed for each gene (Transnetyx, Cordova, TN). C57BL/6 J mice with *Piezo1* gene flanked with LoxP sites (Piezo1$^{flox}$) were described previously (*Li et al., 2014*). To generate tamoxifen (TAM) inducible disruption of *Piezo1* gene in the endothelium *(Piezo1$^{\Delta EC}$)*, Piezo1$^{flox}$ mice were crossed with mice expressing cre recombinase under the Cadherin5 promoter (Tg(Cdh5-cre/ERT2)1Rha) and inbred to obtain Piezo1$^{flox/flox}$/Cdh5-cre mice. TAM (T5648, Sigma-Aldrich, Saint-Louis, MO) was dissolved in corn oil (C8267 Sigma-Aldrich) at 20 mg.ml$^{-1}$. 10–12 week-old male mice were injected intra-peritoneal with 75 mg.kg$^{-1}$ TAM for five consecutive days and studied 10–14 days later. Control mice were the same except they lacked cre, so they could not disrupt Piezo1 even though they were also injected with TAM.

## Acute isolation of liver endothelial cells

Liver of 12–14 week-old male mice was used. Tissue was mechanically separated using forceps, further cut in smaller pieces and incubated at 37°C for 50 min, in a MACSMix Tube Rotator to provide continuous agitation, along with 0.1% Collagenase II (17101–015, Gibco, Waltham, MA) and Dispase Solution (17105–041, Gibco). Following enzymatic digestion samples were passed through 100 μm and 40 μm cell strainers to remove any undigested tissue. The suspension was incubated for 15 min with dead cell removal paramagnetic beads (130-090-101, Miltenyi Biotec GmbH, Bergisch Gladbach, Germany) and then passed through LS column (130-042-401, Miltenyi Biotec). The cell suspension was incubated with CD146 magnetic beads (130-092-007, Miltenyi Biotec 130-092-007) at 4°C for 15 min under continuous agitation and passed through MS column (130-042-201, Miltenyi Biotec). The CD146 positive cells, retained in the MS column, were plunge out with PEB and centrifuged at 1000 RPM for 5 min. Cell pellet was resuspended in RLT buffer (74004, Qiagen, Hilden, Germany) to proceed with RNA isolation.

## Cell culture

HMVEC-Cs were cultured in endothelial medium 2 MV (EGM-2MV, CC-3202, Lonza, Basel, Switzerland) according to the manufacturer's protocol. Sixteen hours before performing experiments, cells were cultured with starvation medium consisting of EGM-2MV but only 0.5% fetal bovine serum and without vascular endothelial growth factor A$_{165}$ (VEGF A$_{165}$) and basic fibroblast growth factor.

## siRNA transfection

HMVEC-Cs were transfected with siRNA using Opti-MEM I Reduced Serum Medium (31985070, ThermoFisher Scientific, Waltham, MA) and Lipofectamine 2000 (11668019, ThermoFisher Scientific). For transfection of cells in 6-well plates, a total of 50 nmol siRNA in 0.1 mL was added to 0.8 mL cell culture medium per well. Medium was changed after 4 hr. After 48 hours cells were exposed to

Yoda1 or SS and subjected to RNA or protein isolation. For $Ca^{2+}$ measurement, cells were plated into a 96-well plate at a density of 25000 cells per well 24 hr after transfection, and $Ca^{2+}$ entry was recorded 24 hr later.

## RNA isolation and RT-qPCR

For isolated liver endothelial cells, RNA was isolated by using RNeasy micro-kit (74004, Qiagen). A total of 100 ng RNA per sample was subjected to Reverse Transcriptase (RT) by using iScript cDNA Synthesis kit (1708890, BioRad, Hercules, CA). For whole liver, RNA was isolated using phenol/chloroform extraction from snap frozen samples. A microgramme of RNA was used for RT (Superscript III Reverse Transcriptase, 18080044, Invitrogen, Carlsbad, CA). qPCR was performed using SyBR Green (1725122, Biorad). The sequences of PCR primers are shown in *Supplementary file 1*. Primers were synthetized by Sigma. qPCR reactions were performed on a LightCycler 480 Real Time PCR System (Roche, Basel, Switzerland). Samples were analysed using the comparative CT method, where fold-change was calculated from the ΔΔCt values with the formula $2^{-\Delta\Delta Ct}$.

## ADAM10 enzyme activity

Activity was determined using the SensoLyte520 ADAM10 Activity Assay Kit (AS-72226, AnaSpec Inc, Fremont, CA), which is based on the FRET substrate 5-FAM/QXL520 with excitation/emission of 490/520 nm. HMVEC-Cs were treated with or without Yoda1 for 30 min in the presence or absence of ADAM10 inhibitor. Cells were then washed with PBS and collected with Trypsin-EDTA. The pellet was resuspended in assay buffer, incubated on ice for 10 min and centrifuged at 10 000 *g* for 10 min at 4°C. The supernatants were plated on a 96-well plate. The substrate solution was diluted in Assay Buffer, brought to 37°C was then mixed 1:1 with the sample. The fluorescence was measured every 2.5 min for 60 min at 37°C at excitation/emission of 490/520 nm Flexstation three microplate reader with SoftMax Pro 5.4.5 software (Molecular Devices, San Josa, CA).

## Shear stress

Endothelial cells were seeded on glass slides (MENSJ5800AMNZ, VWR, Radnor, PA) coated with Fibronectin (F0895, Sigma-Aldrich). Sixteen hours before performing experiments, cells were cultured with starvation medium consisting of EGM-2MV but only 0.5% fetal bovine serum and without vascular endothelial growth factor $A_{165}$ (VEGF $A_{165}$) and basic fibroblast growth factor. The slides were placed in a parallel flow chamber and flow of starvation medium was driven using a peristaltic pump.

## Measurement of intracellular $Ca^{2+}$ concentration ($[Ca^{2+}]_i$)

Cells plated in 96-well plates were incubated for 1 hr in Standard Bath Solution (SBS, containing in mM: 130 NaCl, 5 KCl, 8 D-glucose, 10 HEPES, 1.2 $MgCl_2$, 1.5 $CaCl_2$, pH 7.4) supplemented with 2 µM fura-2-AM (F1201, Molecular Probes, Eugene, OR) and 0.01% pluronic acid. Cells were then washed in SBS at room temperature for 30 min, allowing deesterification to release free fura-2. Fluorescence (F) acquisition (excitation 340 and 380 nm; emission 510 nm) was performed on a Flexstation three microplate reader with SoftMax Pro 5.4.5 software (Molecular Devices). After 60 s of recording, Yoda1 was injected. $Ca^{2+}$ entry was quantified after normalization (ΔF340/380 = F340/380(t)-F340/380(t = 0)).

## Immunoblotting

Proteins were isolated in RIPA buffer supplemented with PMSF, protease inhibitor mixture, and sodium orthovanadate (RIPA Lysis Buffer System, sc24948, Santa Cruz, Dallas, TX). Samples were heated at 95°C for 5 min in SDS-PAGE sample buffer, loaded on a precast 4–20% polyacrylamide gradient gel (4561094, Biorad) and subjected to electrophoresis. Proteins were transferred onto a nitrocellulose membrane (Trans-Blot Turbo RTA Mini Nitrocellulose Transfer Kit, 1704270, BioRad) for 30 min using Trans-Blot Turbo Transfer System (BioRad). Membranes were blocked with 5% milk in Tris-buffered saline with Tween 0.05% for 1 hr at room temperature. The membranes were exposed to primary antibody overnight at 4°C, rinsed and incubated with appropriate horseradish peroxidase-labelled secondary antibody for 1 hr at room temperature. The detection was performed

by using SuperSignal West Femto (34096, ThermoFisher Scientific) and visualized with a G-Box Chemi-XT4 (SynGene, Cambridge, UK). GAPDH was used as reference protein.

## Reagents

Human cardiac microvascular endothelial cells (HMVEC-C, CC-7030, Lonza), DAPT (D5942, Sigma-Aldrich), GI254023X (3995, Tocris Bioscience, Bristol, UK), Yoda1 (5586/10, Tocris Bioscience), ON-TARGET plus Control siRNA (Dharmacon, Lafayette, CO), siRNA Piezo1 (Sigma-Aldrich: 5'- GCAAG UUCGUGCGCGGAUU[dT][dT]- 3'), ON-TARGET plus SMARTpool human siRNA ADAM10 (Dharmacon), cleaved Notch1 Val1744 D3B8 rabbit monoclonal (4147, Cell Signaling Technology, Danvers, MA), rabbit anti-ADAM10 (AB19026, Merck KGaA, Darmstadt, Germany), goat anti human VEGFR2 (AF357, R and D system, Minneapolis, MN), mouse anti-human PECAM-1 (CD31) (M0823, Agilent Dako, Santa Clara, CA), GAPDH mouse anti-human (10R-G109b, Fitzgerald Industries International, Acton, MA) and anti-mouse, anti-rabbit and anti-goat HRP conjugated secondary antibodies (Jackson ImmunoResearch, Ely, UK).

## Statistical analysis

All averaged data are presented as mean ± standard deviation (SD). Statistical significance was determined using two-tailed *t*-test when only two groups were compared or by 2-way ANOVA followed by Tukey posthoc test when multiple groups were treated with vehicle control (DMSO) or Yoda1 were studied. When distribution of data were compared, two-tailed Fisher's exact test was used. The genotypes of mice were blinded to the experimental investigator and studied at random according to Mendelian ratio. In all cases, statistical significance was assumed for probability (*P*) < 0.05. NS indicates when no significant difference was detected. Statistical tests were performed using OriginPro 8.6 software or GraphPad Prism 6.0. The letter n indicates the number of independent biological experiments and its value in each case is stated in figure legends. The number of replicates per independent experiment was one for western blotting, two for qPCR, four for $Ca^{2+}$ assays and one for the ADAM10 activity assay.

## Acknowledgements

The study was supported by EU Marie Skłodowska Curie Individual Fellowship to VC and by grants from the Wellcome Trust and British Heart Foundation to DJB and a Fonds Wetenschappelijk Onderzoek grant G091018N to EAVJ. We thank Richard Cubbon for helpful comments on the manuscript.

## Additional information

### Funding

| Funder | Grant reference number | Author |
| --- | --- | --- |
| Wellcome | 110044/Z/15/Z | David J Beech |
| British Heart Foundation | RG/17/11/33042 | David J Beech |
| European Commission | H2020-MSCA-IF-2016 SAVE 748369 | Vincenza Caolo |
| Fonds Wetenschappelijk Onderzoek | G091018N | Elizabeth AV Jones |

The funders had no role in study design, data collection and interpretation, or the decision to submit the work for publication.

### Author contributions

Vincenza Caolo, David J Beech, Conceptualization, Formal analysis, Supervision, Funding acquisition, Investigation, Methodology, Project administration; Marjolaine Debant, Conceptualization, Formal analysis, Supervision, Funding acquisition, Investigation, Visualization, Methodology; Naima Endesh, Formal analysis, Investigation, Visualization, Methodology; T Simon Futers, Resources, Investigation, Methodology; Laeticia Lichtenstein, Resources, Formal analysis, Investigation, Methodology; Fiona

Bartoli, Formal analysis, Investigation, Methodology; Gregory Parsonage, Investigation, Methodology, Project administration; Elizabeth AV Jones, Conceptualization, Funding acquisition

### Author ORCIDs
Vincenza Caolo https://orcid.org/0000-0001-6215-2702
Marjolaine Debant http://orcid.org/0000-0001-5988-3395
Laeticia Lichtenstein http://orcid.org/0000-0003-3900-786X
David J Beech https://orcid.org/0000-0002-7683-9422

### Ethics
Animal experimentation: All animal use was authorized by the University of Leeds Animal Ethics Committee and Home Office UK (Project Licence P606320FB to David J Beech).

### Decision letter and Author response
Decision letter https://doi.org/10.7554/eLife.50684.sa1
Author response https://doi.org/10.7554/eLife.50684.sa2

## Additional files

### Supplementary files
- Supplementary file 1. PCR primer sequences.
- Transparent reporting form

### Data availability
Source data files have been provided for all 11 data figures and indicated as such in each relevant figure legend.

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
