## [Decision Letter]

**Acceptance summary:**

Notch1 has recently been shown to be activated by fluid shear stress in endothelial cells and to mediate flow-dependent gene regulation and stabilization of the vascular barrier. The mechanisms by which flow induces Notch1 activation have however remained unclear. Caolo et al. now provide data indicating that the mechanosensitive cation channel Piezo1 links fluid shear stress via calcium-dependent regulation of proteinases, to the activation of Notch1. These new findings, contribute to the broad understanding of Notch1 physiology and pathophysiology.

**Decision letter after peer review:**

[Editors’ note: the authors submitted for reconsideration following the decision after peer review. What follows is the decision letter after the first round of review.]

Thank you for submitting your work entitled "Piezo1 channel activates ADAM10 sheddase to regulate Notch1 and gene expression" for consideration by *eLife*. Your article has been reviewed by three peer reviewers, one of whom is a member of our Board of Reviewing Editors, and the evaluation has been overseen by a Senior Editor. The following individual involved in review of your submission has agreed to reveal their identity: Julien Vermot (Reviewer #3).

Having considered your plan for revisions, our decision has been reached after consultation between the editors. Based on these discussions and the individual reviews below, we regret to inform you that your work will not be considered further for publication in *eLife*.

It is our view that the submitted plan does not address at all the main reviewers' comments (that were defined as "essential"), but mainly provides explanations as to why the experiments are not necessary or cannot be performed.

Specifically, in response to point 1, which was one of the major criticisms, the authors do not explain which experiments (if any) they plan to carry out. In response to point 2 and point 3, only a minor experiment would be performed.

Based on the submitted plan, we anticipate that a revised manuscript would not address the essential concerns raised by the reviewers and therefore we have decided against inviting re-submission.

Reviewer #1:

In this study Caolo and co-workers propose a mechanism linking between mechanical forces and Notch1 regulation in endothelial cells, which involves Piezo1 channels and ADAM10 enzymatic activity. Specifically, the authors use the small-molecule Yoda1 to activate Piezo1 channels in HMVECs and assess the cleavage and release of NICD, as well as the levels of expression of downstream Notch1 targets such as Hes1 and Dll4. Using a similar approach, the authors investigate also the potential role of Piezo1 as inducer of ADAM10 enzymatic activity, and show that this step is involved as well in NICD release. Finally, the authors propose that this mechanism is physiologically relevant by showing that liver endothelial cells isolated from Piezo1^∆EC^ mice display specific downregulation of Notch1 target genes.

Overall, the work could be of potential interesting as it identifies a putative mechanistic link between mechanical forces and Notch activation. However, the data come across as being somewhat preliminary and quite speculative. While the presented experiments appear solid in terms of numbers, most assumptions are based solely on a single assay (Yoda1 agonist to activate Piezo1, enzymatic assay to assess ADAM10 activity, etc), making it difficult to reach accurate conclusions. Moreover, some of the results appear over-interpreted and too strong to be drawn from the presented data. For instance, while the authors aim to demonstrate the link between mechanical forces and Notch activation, there's only one experiment that applies actual shear stress, whereas the large majority is based solely on the addition of a small molecule to the medium.

An additional major weakness of this manuscript is that there is a significant imbalance between the characterization of HMVECs and mouse Piezo1^∆EC^ liver endothelial cells. Given that mouse-derived ECs, which provide a more physiological platform, were available to the authors, it is unclear why they didn't use this system to characterize the proposed molecular mechanisms.

Finally, the manuscript is written in a rather superficial way, with no proper elaboration of the observed phenotypes and/or the significance of most of the results. The Introduction and Discussion could be significantly improved by a more detailed, critical, in depth analysis of the findings. Likewise, the Results section will benefit from a more elaborated presentation.

As a whole, this manuscript presents an interesting hypothesis, however it appears to lack the broad scope and impact necessary for publication in *eLife*, and therefore I believe it might suit better a more specialized journal.

Reviewer #2:

Notch1 has recently been shown to be activated by fluid shear stress in endothelial cells and to mediate flow-dependent gene regulation and stabilization of the vascular barrier. The mechanism by which flow induces Notch1 activation has however remained unclear. Caolo et al. now provide data indicating that the mechanosensitive cation channel Piezo1 links fluid shear stress via calcium-dependent regulation of proteinases to the activation of Notch1. This is an exciting new finding, which may be of broader relevance for the understanding of Notch1 physiology and pathophysiology. In several points the study, however, appears a little bit fragmentary and preliminary, and a couple of additional experiments are required to better support the major conclusions of this manuscript.

1) With one exception, all the in vitro data are based on activation of Piezo1 by Yoda1. While Yoda1 appears to be a specific activator of Piezo1, it remains an unphysiological way to stimulate the pathway. It is important that the authors test whether also a physiological endothelial stimulus, such as fluid shear stress, promotes the formation of NICD and the activation of Notch1 downstream gene expression in a Piezo1-dependent manner.

2) The authors use inducible endothelium-specific Piezo1-deficient mice to show that loss of Piezo1 in hepatic sinusoidal endothelial cells results in reduced expression of Notch1-regulated genes. This is an interesting finding but not sufficient to make a strong point as this could also be an indirect phenomenon. It is at least necessary to determine in the Piezo1-deficient liver endothelial cells the formation of NICD and the activity of ADAM10 and/or γ-secretase. To test the physiological significance of this observation, the authors should also consider studying sinusoid development after early postnatal induction of endothelial Piezo1 deficiency or later in adult animals in the context of liver regeneration after e.g. partial hepatectomy.

3) A recently published study describes that shear stress triggers DLL4-dependent proteolytic activation of Notch1 as a mechanism linking fluid shear stress to the stabilization of the endothelial barrier (doi: 10.1038/nature24998). Since this study suggests a somewhat different mechanism of shear stress-induced Notch1 activation, the authors need to discuss this manuscript and test whether Piezo1 mediates e.g. internalization of DLL4 as suggested in this publication.

Reviewer #3:

In this study, Caolo et al. are investigating the mechanism that provides force sensing properties to the notch receptor. The authors provide a nice explanation by implicating piezo1, a well-established stretch sensitive receptor, and a Ca^2+^-regulated transmembrane sheddase that mediates S2 Notch1 cleavage. The results are based on convincing in vitro and in vivo data and are expected to have a significant impact in the field.

I only have a few comments to improve the work:

I have an issue with the timing of the experiments: the authors are assessing fast responses related to piezo and calcium (seconds/minutes) but use tools such as siRNA or conditional KO that takes hours/days to inactivate function. Yoda1 is certainly a convincing way to specifically activate piezo directly but the loss of function approach looks more difficult. Is it possible to use a chemical blocker of piezo, even not extremely specific to piezo, to strengthen this point?

The authors implicate calcium and sodium transients mediated by piezo to explain adam10 activation but never test it. This might not be straightforward but would be a good way to back up the model proposed in the Figure 5.

The choice of 4 notch gene as downstream targets is a bit narrow, as there are certainly much more genes responding to NICD activation. A broader analysis (perhaps mRNAseq) would help to judge better the extent (and specificity) of the misregulation of the notch pathway.

It would be good to control that piezo and ADAM10 are not affecting the cytoskeletal properties of the cells by assessing whether endothelial cells shape and adhesion are normal upon siRNA treatments.

[Editors’ note: further revisions were suggested prior to acceptance, as described below.]

Thank you for choosing to send your work entitled "Shear stress activates ADAM10 sheddase to regulate Notch1 via the Piezo1 force sensor in endothelial cells" for consideration at *eLife*.

The reviewers have looked at the revised version and discussed the reviews with one another. While as stated by the authors the paper is improved, the reviewers are still not fully convinced about the findings and feel that the new data do not fully address their major concerns. Accordingly, the changes summarized below will be essential in order for *eLife* to further consider this manuscript for publication.

1) Testing the effect of Piezo1 knockdown on shear stress induced generation of NICD was an important experiment missing from the manuscript. The authors have not been able to do the experiment in primary cells, however they did it in HMVECs. What they actually found was that both the basal as well as the stimulated level of the NICD was greatly reduced after knockdown of Piezo1, while the shear stress induced increase doesn't appear to be affected (Figure 1A of revised manuscript). This is not really convincing and needs to be further explored.

2) In Figure 1A the authors should provide the ratio of NICD intensity before and after stimulation in both conditions. Looking at the graphs, it seems that this ratio may not be different. This might suggest that piezo known down may have additional effect (such as on cell cytoskeletal properties) in addition to its shear sensing functions as previously suggested by reviewer #3. This point has not been fully addressed in this revised version.

3) It is not clear why the authors were not able to study the mechanism in whole livers. Any data supporting the proposed mechanism on the in vivo level would be absolutely required to strengthen the manuscript.

[Editors' note: further revisions were suggested prior to acceptance, as described below.]

Thank you for submitting your article "Shear stress activates ADAM10 sheddase to regulate Notch1 via the Piezo1 force sensor in endothelial cells" for consideration by *eLife*. Your article has been reviewed by two peer reviewers, and the evaluation has been overseen by a Karina Yaniv as Reviewing Editor and Olga Boudker as the Senior Editor.

The reviewers have discussed the reviews with one another and the Reviewing Editor has drafted this decision to help you prepare a revised submission.

Summary:

Notch1 has recently been shown to be activated by fluid shear stress in endothelial cells and to mediate flow-dependent gene regulation and stabilization of the vascular barrier. The mechanism by which flow induces Notch1 activation has however remained unclear. Caolo et al. now provide data indicating that the mechanosensitive cation channel Piezo1 links fluid shear stress via calcium-dependent regulation of proteinases, to the activation of Notch1. This is an exciting new finding, which may be of broader relevance for the understanding of Notch1 physiology and pathophysiology.

Revisions:

The reviewers were pleased with the way you responded to their concerns. However, they agree that their major comment regarding the in vivo work has not been successfully addressed. In particular, they refer to the inability to see changes in NICD levels in liver endothelial cells lacking Piezo1. While the reviewers understand that this may result from technical reasons, they feel that without these data the manuscript lacks conclusive in vivo evidence for the proposed mechanism linking Piezo1 and expression of Notch1 target genes. Therefore, they request the following changes so that this fact is reflected in the final manuscript:

1) You should add in the scheme (Figure 5) an arrow indicating the possibility of an Adam 10/Notch/NICD independent mechanism

2) You should also include the alternative mechanism of flow-induced Notch signalling described by Polacheck et al., 2017.

---

## [Author Response]

[Editors’ note: The authors appealed the original decision. What follows is the authors’ response to the first round of review.]

Reviewer #1:[…] Overall, the work could be of potential interesting as it identifies a putative mechanistic link between mechanical forces and Notch activation. However, the data come across as being somewhat preliminary and quite speculative. While the presented experiments appear solid in terms of numbers, most assumptions are based solely on a single assay (Yoda1 agonist to activate Piezo1, enzymatic assay to assess ADAM10 activity, etc), making it difficult to reach accurate conclusions. Moreover, some of the results appear over-interpreted and too strong to be drawn from the presented data. For instance, while the authors aim to demonstrate the link between mechanical forces and Notch activation, there's only one experiment that applies actual shear stress, whereas the large majority is based solely on the addition of a small molecule to the medium.

We successfully performed these experiments. The results strongly support relevance of our hypothesis to the physiological stimulus of shear stress.

In the new figure panels we show that shear stress promotes the formation of NICD and the activation of Notch1 downstream gene expression in a Piezo1-dependent manner.

These new data have enabled us to completely reorganise and greatly improve the physiological emphasis of the manuscript. Therefore, the new data are not contained as new stand-alone figures but integrated across the revised figures as a whole.

The original data showing that shear stress stimulates ADAM10 activity in a Piezo1-dependent manner have been retained and integrated in the revised structure of the manuscript.

In addition to the Figures/Legends/Results we revised the Discussion, Abstract and Title to improve the relevance to the physiological stimulus.

An additional major weakness of this manuscript is that there is a significant imbalance between the characterization of HMVECs and mouse Piezo1^∆EC^ liver endothelial cells. Given that mouse-derived ECs, which provide a more physiological platform, were available to the authors, it is unclear why they didn't use this system to characterize the proposed molecular mechanisms.

As indicated above, we increased the physiological relevance of the data through the new shear stress results presented in the revised manuscript. As part of this, we performed deeper investigation of the mouse ECs and the results are positive. We now show that Piezo1 knockout in endothelium affects 8 Notch1-regulated genes: *Hes1*, *Dll4*, *Hey1*, *HeyL*, *Jagged1*, *Hes2*, *Hes3* and *Ephrin B2* (revised Figure 4).

We are not able to repeat all the mechanistic aspects in freshly-isolated ECs. This is because of technical limitations.

We worked to reliably detect NICD in the murine ECs but it has been challenging, we think because of susceptibility of NICD to degradation. The problem is compounded by the need to detect decrease from basal: from low abundance to even lower. In a pilot study we detected basal NICD (full length and degraded) but we only saw what seemed to be degraded NICD when we scaled up for the wildtype and knockout comparison. The data actually support our conclusion that NICD is decreased but we would be concerned about including the data because of the degradation problem. Therefore we are not able to include such data. We don’t yet know how to prevent this degradation, which is not so evident in HMVEC-Cs.

We would probably overcome such a technical challenge by culturing the mouse ECs but this would not add much value to the manuscript because of the similarity to the HMVEC-C approach.

The mechanism can’t be detected in whole liver because it is vascular-specific, as we show in the manuscript. We have worked to develop whole tissue staining of vascular NICD but, so far, it is not sufficiently robust for quantitative data analysis. In our group we have over 10 years of experience with Notch studies, so we don’t think this is because of a lack of our experience or capability, but rather a reflection of limitations of the available Notch1 tools.

We have recognised in the revised Discussion that our conclusions about some aspects of the pathway depend on data from cultured ECs (HMVEC-Cs). We also added recognition of the non-canonical pathway for Notch1 regulation (Polacheck et al., 2017).

Finally, the manuscript is written in a rather superficial way, with no proper elaboration of the observed phenotypes and/or the significance of most of the results. The Introduction and Discussion could be significantly improved by a more detailed, critical, in depth analysis of the findings. Likewise, the Results section will benefit from a more elaborated presentation.

We enhanced the manuscript throughout.

As a whole, this manuscript presents an interesting hypothesis, however it appears to lack the broad scope and impact necessary for publication in eLife, and therefore I believe it might suit better a more specialized journal.Reviewer #2:Notch1 has recently been shown to be activated by fluid shear stress in endothelial cells and to mediate flow-dependent gene regulation and stabilization of the vascular barrier. The mechanism by which flow induces Notch1 activation has however remained unclear. Caolo et al. now provide data indicating that the mechanosensitive cation channel Piezo1 links fluid shear stress via calcium-dependent regulation of proteinases to the activation of Notch1. This is an exciting new finding, which may be of broader relevance for the understanding of Notch1 physiology and pathophysiology. In several points the study, however, appears a little bit fragmentary and preliminary, and a couple of additional experiments are required to better support the major conclusions of this manuscript.1) With one exception, all the in vitro data are based on activation of Piezo1 by Yoda1. While Yoda1 appears to be a specific activator of Piezo1, it remains an unphysiological way to stimulate the pathway. It is important that the authors test whether also a physiological endothelial stimulus, such as fluid shear stress, promotes the formation of NICD and the activation of Notch1 downstream gene expression in a Piezo1-dependent manner.

We successfully performed these experiments. The results strongly support relevance of our hypothesis to the physiological stimulus of shear stress.

In the new figure panels we show that shear stress promotes the formation of NICD and the activation of Notch1 downstream gene expression in a Piezo1-dependent manner.

These new data have enabled us to completely reorganise and greatly improve the physiological emphasis of the manuscript. Therefore, the new data are not contained as new stand-alone figures but integrated across the revised figures as a whole.

The original data showing that shear stress stimulates ADAM10 activity in a Piezo1-dependent manner have been retained and integrated in the revised structure of the manuscript.

In addition to the Figures/Legends/Results we revised the Discussion, Abstract and Title to improve the relevance to the physiological stimulus.

2) The authors use inducible endothelium-specific Piezo1-deficient mice to show that loss of Piezo1 in hepatic sinusoidal endothelial cells results in reduced expression of Notch1-regulated genes. This is an interesting finding but not sufficient to make a strong point as this could also be an indirect phenomenon. It is at least necessary to determine in the Piezo1-deficient liver endothelial cells the formation of NICD and the activity of ADAM10 and/or γ-secretase. To test the physiological significance of this observation, the authors should also consider studying sinusoid development after early postnatal induction of endothelial Piezo1 deficiency or later in adult animals in the context of liver regeneration after e.g. partial hepatectomy.

As indicated above, we increased the physiological relevance of the data through the new shear stress results presented in the revised manuscript. As part of this, we performed deeper investigation of the mouse ECs and the results are positive. We now show that Piezo1 knockout in endothelium affects 8 Notch1-regulated genes: *Hes1*, *Dll4*, *Hey1*, *HeyL*, *Jagged1*, *Hes2*, *Hes3* and *Ephrin B2* (revised Figure 4).

We are not able to repeat all the mechanistic aspects in freshly-isolated ECs. This is because of technical limitations.

We worked to reliably detect NICD in the murine ECs but it has been challenging, we think because of unexpected susceptibility of NICD to degradation. The problem is compounded by the need to detect a decrease from basal: from low abundance to even lower. In a pilot study we detected basal NICD (full length and degraded) but we have only seen what seems to be degraded NICD when we scaled up for the wildtype and knockout comparison. The data actually support our conclusion that NICD is decreased but we would be concerned about including the data because of the degradation. Therefore we are not able to include such data at the moment.

We would likely overcome such a technical challenge by culturing the mouse ECs but this would not add much value to the manuscript because of the similarity to the HMVEC-C approach.

The mechanism can’t be detected in whole liver because it is vascular-specific, as we show in the manuscript. We have worked to develop whole tissue staining of vascular NICD but, so far, it is not sufficiently robust for quantitative data analysis. In our group we have over 10 years of experience with Notch studies, so we don’t think this is because of a lack of our experience or capability, but rather a reflection of limitations of the available Notch1 tools.

It is questionable whether in vivo-relevant enzyme activity can be preserved during cell isolation and, to the best of our knowledge, there is no methodology for detection of ADAM10 activity in tissue sections.

We have recognised in the revised Discussion that our conclusions about some aspects of the pathway depend on data from cultured ECs (HMVEC-Cs).

Prior work has shown that endothelial Piezo1 is important for vascular development and structure. While we don’t show such changes specifically for the liver, it is expected that they will be seen in endothelial Piezo1 knockouts if studied at suitable time-points. However, such studies will not necessarily implicate Piezo1 signalling to Notch1 because Piezo1 regulates multiple distinct downstream pathways (Beech and Kalli, 2019). Future studies may be able to reveal how specific disruption of Piezo1 signalling to Notch1 can be achieved in vivo. We addressed these topics in the revised Introduction and Discussion.

3) A recently published study describes that shear stress triggers DLL4-dependent proteolytic activation of Notch1 as a mechanism linking fluid shear stress to the stabilization of the endothelial barrier (doi: 10.1038/nature24998). Since this study suggests a somewhat different mechanism of shear stress-induced Notch1 activation, the authors need to discuss this manuscript and test whether Piezo1 mediates e.g. internalization of DLL4 as suggested in this publication.

Our manuscript is about activation of canonical Notch1 regulation by Piezo1 as a mechanism for it to achieve force sensitivity. We do not exclude other pathways and recognise this in the revised manuscript. We added Discussion of the proposed non-canonical pathway for Notch1 regulation (Polacheck et al., 2017).

Reviewer #3:I only have a few comments to improve the work:I have an issue with the timing of the experiments: the authors are assessing fast responses related to piezo and calcium (seconds/minutes) but use tools such as siRNA or conditional KO that takes hours/days to inactivate function. Yoda1 is certainly a convincing way to specifically activate piezo directly but the loss of function approach looks more difficult. Is it possible to use a chemical blocker of piezo, even not extremely specific to piezo, to strengthen this point?

Unfortunately, the known chemical blockers of Piezo are very limited and lack specificity. We worked to address this limitation through recent screening of a library of more than 10,000 small-molecules against Piezo1. The work is on-going and not yet available to benefit studies such as the one described here.

Nevertheless, we have been able to show that Gd3+, a Piezo1 blocker (Coste et al., 2010), prevents Yoda1 from stimulating ADAM10 activation. We have added a specific section in the Results. The result suggests that it is indeed the fast Piezo1 ion-permeation mechanism that is responsible for the ADAM10/Notch1 activation. We speculate that effects such as ADAM10 activation and NICD formation are triggered quickly but that it then takes time for sufficient accumulation that is detectable by available assays, which are relatively insensitive. We can’t prove this, however, and don’t exclude other intermediates between Piezo1 and ADAM10. We added a paragraph on this topic to the Discussion.

The authors implicate calcium and sodium transients mediated by piezo to explain adam10 activation but never test it. This might not be straightforward but would be a good way to back up the model proposed in the Figure 5.

Testing the roles of Ca^2+^ or Na^+^ might be attempted by removing these ions from the extracellular medium, but this approach is problematic because these ions are very important for so many cellular processes. Such data are of questionable value. In place of this we favoured the Gd3+ approach, as indicated above. This has succeeded and supports the idea that cation flux is important – possibly Ca^2+^. The new paragraph in the Discussion, as indicated above, addresses this point.

The choice of 4 notch gene as downstream targets is a bit narrow, as there are certainly much more genes responding to NICD activation. A broader analysis (perhaps mRNAseq) would help to judge better the extent (and specificity) of the misregulation of the notch pathway.

We now more than doubled the number of genes tested for culture cells and freshly-isolated ECs from the mice. The new data are included in the revised / new figures. The data suggest a broad effect on these genes, especially in vivo (Figure 4).

Piezo1 regulates multiple distinct downstream pathways (Beech and Kalli, 2019). Therefore, RNAseq for wildtype vs Piezo1 knockouts will not yield information specific to the Notch hypothesis. We addressed this topic in the Discussion.

We plan studies to understand how specific disruption of Piezo1/ADAM10/Notch1 might be achieved in vivo. If successful, we would then expect to do RNAseq to determine any wider implications of this pathway.

It would be good to control that piezo and ADAM10 are not affecting the cytoskeletal properties of the cells by assessing whether endothelial cells shape and adhesion are normal upon siRNA treatments.

We previously published on the effect of Piezo1 siRNA on cytoskeletal properties in cultured endothelial cells (Li et al., 2014). There is tightening of plasma membrane-associated actin cytoskeleton and resistance of the cells to alignment in the direction of shear stress. The cells adhered to the substrate in culture and we routinely study endothelial cells from Piezo1 knockouts that adhere to the substrate, but we have seen altered adherence and it has been reported by others (Nonomura et al., 2018).

Piezo1 regulates multiple distinct downstream pathways, as reviewed recently by us (Beech and Kalli, 2019). We are not suggesting there is specific effect only on Notch1 signalling. We added more text to make this clearer in the Discussion.

[Editors’ note: what follows is the authors’ response to the second round of review.]

The reviewers have looked at the revised version and discussed the reviews with one another. While as stated by the authors the paper is improved, the reviewers are still not fully convinced about the findings and feel that the new data do not fully address their major concerns. Accordingly, the changes summarized below will be essential in order for eLife to further consider this manuscript for publication.1) Testing the effect of Piezo1 knockdown on shear stress induced generation of NICD was an important experiment missing from the manuscript. The authors have not been able to do the experiment in primary cells, however they did it in HMVECs. What they actually found was that both the basal as well as the stimulated level of the NICD was greatly reduced after knockdown of Piezo1, while the shear stress induced increase doesn't appear to be affected (Figure 1A of revised manuscript). This is not really convincing and needs to be further explored.

Thank you for drawing attention to this important aspect. We had been too concise and not properly explained the data. Moreover, the example raw data of Figure 1A were not representative of the statistical outcome of the experiments and so we changed to another example.

The statistical outcome of the experiments is that stimulated NICD was reduced to the control level, whereas basal NICD was not affected and shear stress (SS) did not increase NICD above the basal level in the Piezo1 knockdown (siPiezo1) group (Figure 1B). We added “NS” (Not Significant) to Figure 1B to clarify the statistical outcome. Therefore, the concerns raised are not correct according to the statistical analysis. Nevertheless, we understand why there was concern and accordingly changed Figure 1A to another example, revised the text to help understanding and better recognised the complexity and potential limitations of the underlying data through addition of discussion.

More repetitions of the experiment could be done if these changes were felt to be unsatisfactory, but power calculation analysis using the known variance and anticipated effect size indicates that n=17 independent repeats per group would be required to achieve statistical significance for an effect of shear stress after knockdown of Piezo1. This implies that there is an effect to be detected, but one that is difficult to prove under these conditions. Such an n value would be much in excess of norms for this type of labourintensive and time-consuming experiment and could be construed as biased in favour of detecting an effect when a relatively small n value was sufficient to demonstrate significance in the control group.

We had previously used the absence of asterisks to indicate no significant difference but addition of “NS” makes it clearer. We therefore now inserted NS throughout all 11 data figures.

Below we briefly consider the issues regarding Figure 1A, B:

Knockdown of Piezo1 strongly reduced the amount of NICD in the Shear Stress (SS) condition, in fact so much so that it became similar to the Static condition (siCtrl). This result is seen in the example data of Figure 1A and quantitative data of Figure 1B, in which “SS siPiezo1” was not significantly (“NS”) different from “Static siCtrl”; whereas “SS siPiezo1” was significantly less than SS siCtrl (** P < 0.01). We added “NS” to Figure 1B to clarify.

From the quantitative data of Figure 1B we must conclude that knockdown of Piezo1 had no effect on basal NICD in these experiments. We have clarified by adding “NS” but we recognise that there is an appearance of decreased basal NICD in the mean data and such a decrease is evident in the individual experiment shown in Figure 1A (“Static siPiezo1” *cf* “Static siCtrl”). Moreover, related quantitative data of Figure 1D show significant effect on basal NICD (Figure 1D, “DMSO siPiezo1” *cf* “DMSO siCtrl”), apparently contradicting Figure 1B. The experiments of Figure 1B and Figure 1D were, however, not exactly the same. Most obviously the experiments of Figure 1d included DMSO, but Figure 1B was also done in a separate special chamber to enable shear stress studies. We suggest, therefore, that the ability of Piezo1 to regulate basal NICD is particularly sensitive to the cell environment (more sensitive than the effect of Piezo1 to enable induction of NICD by Shear Stress). We have added discussion of this topic.

We appreciate that any effect of treatment on basal NICD creates a challenge for determining the effect of shear stress on top of this because the reference point (basal NICD) is (or may be) different. We added discussion of this matter to better recognise it, explain that it is a potential limitation of the study and to suggest interpretations (e.g. about why there might be a basal effect and why it might be variably observed).

Importantly, the independent data of Figure 4 (B, C) cleanly demonstrate Piezo1-dependence of shear stress effects on Notch1-target gene activation without complication from basal effects.

We accept that the data are not as simple as might be hoped because of the issues around the basal signal. It is clear throughout the study that the topic of basal NICD regulation is an unexpected, interesting and relevant one, as is the topic of the effect of shear stress on top of the basal signal.

To the Results we added and revised as follows:

“Strikingly, depletion of Piezo1 strongly reduced the amount of NICD in the shear stress condition, so much so that it became similar to that of the static control siRNA condition (Figure 1A, B). […] The data suggest that Piezo1 is needed for normal elevation of NICD in shear stress and that it may be a factor regulating basal NICD.”

To the Discussion we added and revised as follows:

“A challenging aspect of our HMVEC-C experiments was the tendency for the inhibition or depletion of ADAM10/Piezo1 to reduce the constitutive abundance of NICD. […] We previously showed that Piezo1 depletion affects cytoskeletal structure in endothelial cells (9) and so there is a possibility that this plays a role in the interplay between Piezo1 and Notch1.”

2) In Figure 1A the authors should provide the ratio of NICD intensity before and after stimulation in both conditions. Looking at the graphs, it seems that this ratio may not be different. This might suggest that piezo known down may have additional effect (such as on cell cytoskeletal properties) in addition to its shear sensing functions as previously suggested by reviewer #3. This point has not been fully addressed in this revised version.

The first part of this point is closely related to point 1, to which we responded above.

If we normalise the “Static siPiezo1” data of the Figure 1B to 1.0, the “SS siPiezo1” NICD data appear almost two times larger (to be precise, 1.85 times). As you have identified, this is already apparent in the original presentation of Figure 1B. It remains the same in the revised version. It is important to note that changing the presentation does not change the fact that there was no significant difference. We have correctly reached conclusions based on the results of the statistical tests but now provide better explanation and discussion to help with the situation, as described in response to point 1.

We draw attention to the raw data of Figure 1A where the basal NICD signal in “Static siPiezo1” was seen to be weak and only just above background. Using such data as a reference, or normalisation, point is precarious and leads to large variation in the associated data. Moreover, it can be seen that the NICD signal was generally weak in the siPiezo1 group, which we think explains, at least in part, the high variance of these data points. We also added discussion about this point to the main manuscript, as described above.

Despite these difficulties, it is clear that the total amount of NICD achievable in the Shear Stress condition was much less after Piezo1 knockdown, and validated strongly by objective statistical analysis.

The second part of this point relates to additional effects of Piezo1 knockdown and potential roles of the cell cytoskeleton. We completely agree that Piezo1 has multiple effects and we worked to make this clearer in the Discussion. Our group has provided some of the evidence in the literature and we recently comprehensively reviewed all of the available evidence – we cite this work and other work. We now further enhanced the discussion on these points by providing more details on the cytoskeleton, referring to possible roles of Piezo1 in addition to shear stress sensing and developing a hypothesis for how Piezo1’s relationship to cytoskeleton might be an explanation for regulation of basal NICD.

We added discussion as follows:

“There are several possible explanations for a basal effect on NICD because multiple signalling pathways are activated by Piezo1. […] We previously showed that Piezo1 depletion affects cytoskeletal structure in endothelial cells (9) and so there is a possibility that this plays a role in the interplay between Piezo1 and Notch1.”

3) It is not clear why the authors were not able to study the mechanism in whole livers. Any data supporting the proposed mechanism on the in vivo level would be absolutely required to strengthen the manuscript.

We apologise that this wasn’t sufficiently clear. We worked to improve this section. Such data are in the section entitled “Endothelial Piezo1 is required for Notch1 target gene expression in mice” and associated Figure 4 and Figure 4—figure supplements 2 and 3. There was no change in Notch1-target gene expression in whole liver of endothelial-specific Piezo1 knockouts (Figure 4—figure supplements 2 and 3). Endothelium is a minor fraction of whole liver. When we separated the endothelium (without cell culture), we saw clear effects on Notch1-target genes (Figures 4). Therefore, we show data supporting the proposed mechanism at the in vivo level; i.e., using knockout mice and the study of freshly-isolated endothelium from whole liver.

We have worked to achieve NICD-specific staining in tissue sections but the quality of NICD antibodies is limited, especially for detecting a decrease of NICD below basal levels (the expected effect of Piezo1 knockout). We have added discussion to recognise this limitation.

Piezo1 signals to multiple pathways, not only ADAM10 and Notch1. Therefore, studies in search of a liver phenotype of Piezo1 knockouts will not specifically address the Piezo1ADAM10-Notch1 hypothesis and would cause a loss of focus that could be misleading by implying an in vivo role of Piezo1-ADAM10-Notch1 signalling that might actually be due to some other function of Piezo1.

We revised and enhanced the Discussion as follows:

“The specific downstream implications of Piezo1-ADAM10/Notch1 signalling in the liver remain to be explicitly determined but we know that prior work showed important effects of endothelial-specific disruption of Notch1-regulated RBPJ on hepatic microvasculature. […] We will be able to determine the specific significance in vivo only if we first determine how to specifically disrupt this pathway relative to others.”

If there are further experiments you would like us to consider, we would be grateful if you would take into account that the Covid-19 outbreak has caused our laboratories to be temporarily shut down and many of our animals to be culled. Significant time, planning and new resource would be needed to provide new in vivo data. Limits on current funding and contracts might prevent such work if our campus shutdown continues for an extended period.

[Editors' note: further revisions were suggested prior to acceptance, as described below.]

Revisions:The reviewers were pleased with the way you responded to their concerns. However, they agree that their major comment regarding the in vivo work has not been successfully addressed. In particular, they refer to the inability to see changes in NICD levels in liver endothelial cells lacking Piezo1. While the reviewers understand that this may result from technical reasons, they feel that without these data the manuscript lacks conclusive in vivo evidence for the proposed mechanism linking Piezo1 and expression of Notch1 target genes. Therefore, they request the following changes so that this fact is reflected in the final manuscript:1) You should add in the scheme (Figure 5) an arrow indicating the possibility of an Adam 10/Notch/NICD independent mechanism2) You should also include the alternative mechanism of flow-induced Notch signalling described by Polacheck et al., 2017.

We made the requested changes to Figure 5. We accordingly added this sentence to the Figure 5 legend:

“In the schematic we also include the suggested contributions from ADAM10/Notch1/NICD independent signalling from Piezo1 and non-canonical Notch1 signalling via the Notch1 transmembrane domain (TMD), as described and referenced in the text.”

We reference and describe the Polacheck work in the manuscript.